# Position: Quantum Program Generation Must Prioritize Validity Over Probabilistic Scaling

**Junhao Song** [* 1]  **Yu Zhou** [* 2]  **William Knottenbelt** [1]  **Yudong Cao** [3 4]

## Abstract

The scaling hypothesis assumes that increasing model parameters yields emergent reasoning capabilities. This position paper argues that applying this probabilistic paradigm to generic quantum circuit synthesis is a directional error. Unlike natural languages, quantum circuits require strict adherence to mathematical constraints that manifest a significant syntax-semantics gap. Training on unverified quantum programs means that models learn syntax but fail to capture the physical semantics of the Hilbert space. Since the valid subset of circuit designs decays exponentially with the number of qubits, post-hoc filtering is mathematically intractable. We propose a pivot from human-centric copilots to verifier-centric agents. We integrate hierarchical constraints, topological masks, and symbolic proxies directly into generation. Our analysis suggests that scale alone cannot bridge the validity gap. Verification-aware architectures offer a viable path for modular quantum program generation. These considerations point toward generation methods that encode task-specific rules of quantum information, rather than relying on imitation alone.

## 1. Introduction

Machine learning research has flourished often with the conflation of fluency with validity in sequence generation tasks. A syntactically imperfect function generated by large language models (LLMs) often retains semantic utility (Austin et al., 2021). Conversely, a syntactically correct statement may be semantically wrong, but may still be an informative data point for model training (Chen et al., 2021). In classical software engineering, this conflation is forgiving. An off-by-one error in a Python script is a localized failure; it is often debuggable via simple runtime execution. For quantum computing, however, it is dangerous to inherit this "copilot" tradition. It blindly imports the assumption that partial correctness yields partial utility. In the quantum realm, a misplaced operation is not merely a bug. Depending on its position, it can invalidate the entire circuit design. A single misplaced gate does not just introduce a localized error. It can destroy the global interference pattern required for computation. Figure 1 contrasts this naive closed-loop paradigm with a verifier-centric alternative, clarifying that the core failure lies not in the absence of verification, but in its post-hoc and non-constructive use.

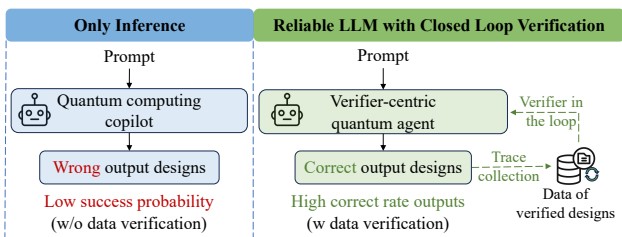

*Figure 1.* **Comparison of the traditional method and reliable LLM with closed loop verification.** When data verification is integrated into the training loop, only validated designs from the constrained validation space are used, ensuring higher correct rate outputs.

Recent systems like IBM's Qiskit Code Assistant (Dupuis et al., 2024) and DeepMind's AlphaTensor-Quantum (Ruiz et al., 2025) show promise. However, a critical dissection reveals their success stems from domain-specific verification loops (Romera-Paredes et al., 2024) or curated human-verified datasets (Vishwakarma et al., 2024). It does not stem from raw parameter scaling. Outside these verified islands, the "scaling hypothesis" (Kaplan et al., 2020) faces a hard mathematical wall. Standard LLMs maximize the likelihood of the next token. They do not maximize the fidelity of the resulting quantum state. There is a fundamental misalignment between the probabilistic objective of the model and the deterministic constraints of the Hilbert space.

---

[*]Equal contribution  [1]Department of Computing, Imperial College London, London, United Kingdom [2]Department of Earth Science and Engineering, Imperial College London, London, United Kingdom [3]Zapata Quantum, Boston, Massachusetts, United States of America [4]BCG X AI Science Institute, Boston Consulting Group, Boston, United States of America. Correspondence to: William Knottenbelt <w.knottenbelt@imperial.ac.uk>, Yudong Cao <ycao@zapataquantum.com>.

*Proceedings of the 43rd International Conference on Machine Learning*, Seoul, South Korea. PMLR 306, 2026. Copyright 2026 by the author(s).

The magnitude of this misalignment is often underestimated. The unitary group $SU(2^n)$ is a continuous manifold of dimension $4^n - 1$ (Nielsen & Chuang, 2010), whereas any polynomial-depth circuit ansatz spans only an $O(\text{poly}(n))$-dimensional submanifold. As a consequence, the set of circuit designs implementing a given target unitary occupies an exponentially sparse subset of the space of all $n$-qubit circuits (Mele, 2024), a fact we formalize via Haar-measure concentration in Section 2.1. For generic synthesis tasks, valid outcomes therefore constitute a vanishingly small fraction of the search space (Bouland et al., 2018). In this regime, scaling computation does not bridge the entropy gap. It merely allows the model to hallucinate more convincingly. A larger model simply learns to generate more sophisticated noise.

> We argue that quantum program generation and other similar scientific domains with hard physical and mathematical constraints expose a fundamental syntax–semantics gap, where applying probabilistic scaling of large models can be entirely counterproductive. The exponential sparsity of functionally correct outputs makes post-selection intractable, so reliability must come from a constructively verified system.

To transition from probabilistic mimicry to scientific discovery, we formalize three critical shifts:

1. **Data poisoning (Section 2):** Training on unverified open-source code introduces a structural bias against physical validity, or data poisoning in the broadest sense of a corrupted training signal. Public quantum code carries no correctness guarantee: parseable code need not implement a valid circuit. Models trained on this distribution learn syntax but fail to capture Hilbert space semantics. We hypothesize that this leads to **inverse scaling** (McKenzie et al., 2023). Larger models more faithfully imitate the biased distribution, becoming more confident in physically invalid outputs.
2. **The complexity trap (Section 3):** We demonstrate that the sparsity of valid circuits renders post-hoc filtering computationally intractable for generic synthesis. Beyond approximately 40-50 qubits, classical verification becomes computationally intractable (Dalzell et al., 2020; Arute et al., 2019). This renders post-hoc filtering infeasible. The exponential decay of the valid subspace creates a coupled barrier that scaling model parameters alone cannot overcome.
3. **Verifier-centric agents (Section 4):** Human-in-the-loop verification is ill-suited for this domain. Human verification is cognitively impractical in the regime of a large number of qubits. We propose replacing human supervisors with formal solvers. We introduce constructive verification protocols. These integrate hierarchical constraints (topological masks, symbolic heuristics, and modular simulation) along with trace-based corpora to enable validity enforcement directly during generation.

## 2. The Structural Limitations of Probabilistic Scaling

The prevailing scaling hypothesis implicitly assumes that increasing model parameters yields emergent reasoning capabilities (Kaplan et al., 2020; Hoffmann et al., 2022). While empirically validated in loosely constrained domains like natural language, this paradigm encounters fundamental difficulties in quantum program generation. We argue that applying probabilistic scaling to arbitrary quantum circuit design constitutes a structural mismatch that cannot be resolved by scaling alone.

### 2.1. The Syntax-semantics Gap

In classical programming, the syntax–semantics gap is often narrow in practice because semantic correctness can be cheaply checked through execution, tests, debuggers and introspection, while syntax errors are frequently localized and repairable (Figure 2). For instance, it has been shown that automated recovery can repair the majority of real-world syntactically invalid programs within seconds (Diekmann & Tratt, 2020), which is evidence that many "syntactically imperfect" programs still preserve enough structural intent to be repaired into a meaningful artifact. We illustrate this fundamental difference with a detailed example in Section A of the Appendix.

Recent AI4Science results indicate that when outputs must satisfy hard mathematical constraints, LLMs behave primarily as high-recall proposal generators rather than reliable constrained designers: their outputs often appear plausible yet fail formal checks, and performance improves mainly when an external verifier or solver is integrated into the loop. In planning, it is shown that autoregressive LLMs are unreliable self-verifiers, motivating LLM-Modulo architectures where model-based verifiers enforce constraints and guide iterative repair (Kambhampati et al., 2024). In formal mathematics, evaluations of Lean4 autoformalization reveal persistent failures on harder theorems (Gulati et al., 2024), while APOLLO demonstrates that substantial gains arise from compiler-guided proof repair rather than raw generation (Ospanov et al., 2025). Together, these findings support the quantum-relevant claim that LLMs struggle in mathematically constrained domains unless generation is tightly coupled with explicit checking and repair.

A second, independent limitation arises in domains where the true optimization objective only becomes visible after crossing abstraction layers i.e., leaky abstraction. In such settings, local high-level proxies are unreliable because

downstream compilation, synthesis, and optimization introduce nonlocal interactions and heuristic effects. Results in similarly structured domains such compiler optimization (Cummins et al., 2023), hardware design (Fang et al., 2024) also support the claim that LLMs struggle with leaky-abstraction design problems. In the case of quantum compilation (Cuccaro et al., 2004; Javadi-Abhari et al., 2024), even among valid designs, locally "better-looking" structures may invert after transpilation, routing, or scheduling, necessitating multi-level evaluation rather than syntax-level heuristics. The model can be simultaneously (i) confidently fluent while violating hard physical/structural constraints, and (ii) confidently suboptimal because the meaning of a design decision only materializes after lower-level compilation, so without explicit verification and multi-level evaluation in the loop, MLE-trained fluency systematically diverges from both constraint satisfiability and compiled quality.

> **BACKGROUND 1:** *Syntax–semantics gap* : In quantum programming, these "syntax-first" conveniences break down, and the syntax–semantics gap widens for two independent reasons:
>
> - Semantic inspection is intrinsically constrained: common debugging practices such as inspecting intermediate states or step-by-step execution are obstructed because quantum states in general are hard to capture exactly with only classical resources, and on a quantum device, measurement collapses quantum states. In addition, the no-cloning theorem prevents faithful copying, making even the validation of a syntactically correct circuit non-trivial (Ramalho et al., 2024).
> - Even among semantically correct programs, cost semantics leak across abstraction layers, rendering local design judgments unreliable. For example, the ripple-carry adder (Cuccaro et al., 2004) admits a clean modular description in terms of MAJ and UMA blocks (see Figure 5a), yet apparent module-level improvements (e.g., fewer gates) may invert after gate-level lowering and compiler rewrites such as cancellation, resynthesis, scheduling, and routing. The true optimization objectives are often determined by cross-module interactions that are not apparent at the modular level (Javadi-Abhari et al., 2024). This is a clear case of abstraction leakage, which we discuss in detail in Section 4.2.

## 2.2. Structural Bias from Unverified Corpora

Training on even lightly curated quantum code biases the learned distribution against physical validity, in a manner structurally analogous to data poisoning (Tsai et al., 2025). Indeed, practitioners building production quantum-code LLMs already recognize that uncurated public code is unsuitable as training data. In the Qiskit Code Assistant train-

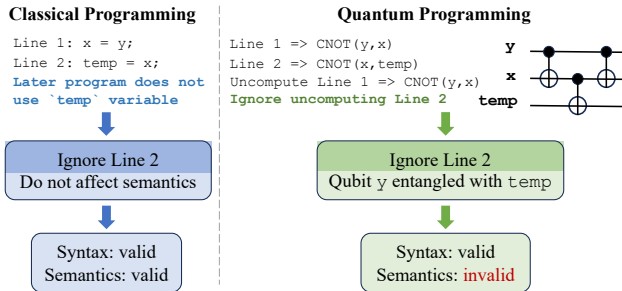

*Figure 2.* **Toy example contrasting classical and quantum programming:** syntactic fluency is much less of a predictor of semantic correctness in quantum programming than in classical programming.

ing pipeline (Dupuis et al., 2024), code from official Qiskit GitHub organizations is oversampled by $10.3\times$ relative to general scraped repositories, a ratio chosen empirically for benchmark performance. We take such filtering as a starting point and consider whether the curation strategies currently in use can close the validity gap identified in Section 2.1.

Source-based heuristics for curating code data select code that are apparently well-tested, but not code that is guaranteed to implement the intended function. They work in classical settings for an indirect reason: classical code is constantly tested in practice, through unit tests, CI, crashes, and bug reports, so incorrect code gets caught and fixed over time, and high-provenance repositories end up being more trustworthy as a result. Quantum code lacks this tight feedback loop. Incorrect programs are not pruned from public corpora, so filtering by source does little to guarantee correctness, and training on the resulting data carries that gap into the model. Our earlier work (Song et al., 2026) reports empirical evidence consistent with this hypothesis: LLMs trained without data verification ceil out at $79\%$ accuracy on quantum circuit optimization tasks, with no further gains from scale.

## 2.3. The Phenomenon of Inverse Scaling

We hypothesize that quantum program generation exhibits a form of inverse scaling (McKenzie et al., 2023): a regime in which improvements in next-token loss do not translate into, and can actively diverge from, improvements in physical validity. Standard scaling laws predict that larger models achieve lower next-token loss on the training distribution (Kaplan et al., 2020). In our domain, however, the training distribution itself is biased away from the manifold of valid programs (Section 2.2), so models that fit it more faithfully at scale fit a biased target more faithfully.

This adaptation manifests at both training and inference time. At training, under maximum likelihood on a biased corpus, the model's distribution $p_\theta$ converges toward the

training distribution $p_{\text{train}}$ rather than toward the distribution $p_{\text{valid}}$ of physically valid circuits. Since $p_{\text{train}}$ is structurally misaligned with $p_{\text{valid}}$, the KL divergence $D_{KL}(p_{\text{valid}} || p_\theta)$ does not vanish with scale: probability mass concentrates increasingly tightly on syntactically plausible but physically invalid regions of the circuit space. At inference, larger models acquire stronger syntactic priors that are correspondingly harder to override. The Strong Prior modality (McKenzie et al., 2023), which is also demonstrated in classical code (Miceli-Barone et al., 2023), in which large models fail to follow in-context instructions that conflict with patterns learned in pretraining.

> **POSITION 1:** We contend that the objective of generative models in quantum computing must shift from *probabilistic emulation* of program code to *constructive verification* against the mathematical structure of quantum algorithms and subroutines. Scale alone, without verification-aware constraints, cannot bridge the gap between syntax and semantics.

> **BACKGROUND 2:** *Validity* : In our context we are concerned with validity of a quantum program on two distinct logical layers:
> - Structural validity ($\mathcal{M}_{\text{struct}}$): Respecting the constraints imposed by the underlying hardware abstraction. For Noisy Intermediate Scale Quantum (NISQ) devices, this reduces to a coupling graph check on physical gates, which is polynomial time in circuit size. For fault-tolerant quantum computing (FTQC) architectures, this expands to a hierarchy of constraints including patch-layout adjacency and boundary-type matching for lattice surgery (Litinski, 2019; Fowler & Gidney, 2018), magic-state factory throughput and scheduling, code-distance budgeting, and Pauli-frame consistency, several of which are individually NP-hard as combinatorial problems.
> - Functional correctness ($\mathcal{M}_{\text{func}}$): Implementing the target logical channel within tolerance $\epsilon$, namely the circuit's action on the algorithm's computational register matches some $U_{\text{target}} \in SU(2^n)$, where $n$ is the logical qubit count. This definition applies to both NISQ and FTQC regimes. Exact verification requires $\mathcal{O}(2^n)$ time in the worst case. Even approximate decision versions such as the non-identity testing (deciding whether a given unitary circuit is close, up to global phase, to the identity of quantum circuits), are known to be computationally hard (Watrous, 2009). Physical-level concerns specific to fault tolerance, such as decoder convergence and magic-state fidelity, are not in scope for this correctness check.

## 3. The Challenge of Post-selection

Proponents of scaling argue that hallucination can be mitigated through sample-and-filter pipelines. This "Generate-then-Verify" paradigm assumes one can generate $K$ candidates, discard invalid ones and recover validity in expectation as $K$ grows (Li et al., 2022). While effective for domains with locally checkable constraints such as parser-enforceable syntax (Shin et al., 2021) or unit-test filtering (Chen et al., 2021; Li et al., 2022), this strategy fails for generic quantum synthesis. The verification step can be itself computationally expensive, and only an exponentially small fraction of candidates passes. The sample-and-filter pipeline therefore inherits a coupled exponential cost that no amount of model scaling can amortize away.

Validity for quantum synthesis decomposes into two layers (Background 2): structural validity ($\mathcal{M}_{\text{struct}}$), which captures whether a candidate respects the architectural rules of the target hardware abstraction, and functional correctness ($\mathcal{M}_{\text{func}}$), which captures whether the circuit's logical action matches the target unitary. The overall success rate of a sample-and-filter pipeline is the joint probability $P(\mathcal{M}_{\text{struct}}) \, P(\mathcal{M}_{\text{func}} \mid \mathcal{M}_{\text{struct}})$, and the per-candidate verification cost is dominated by the cost of deciding membership in $\mathcal{M}_{\text{func}}$.

### 3.1. Structural Validity: Cost of Checking and Sampling

We analyze $\mathcal{M}_{\text{struct}}$ along two axes: the per-candidate cost of *verifying* membership, and the probability that a candidate drawn from a model's distribution $\pi_\theta$ *satisfies* it. The two axes scale very differently across regimes. Verification is uniformly cheap: each constraint defining $\mathcal{M}_{\text{struct}}$ is local in the candidate description $x$, so for a circuit of $n$ qubits at depth $d$ with $G = \Theta(nd)$ operations, membership can be decided in $\Theta(nd)$ time by per-operation table lookups, gate-set and coupling-graph adjacency in NISQ, and patch-layout, boundary-type, distance-budget, factory-throughput, and Pauli-frame consistency in FTQC (Litinski, 2019; Beverland et al., 2022). The lower bound is immediate by the need to read each constrained operation at least once. We treat layout and schedule annotations as part of $x$: this scopes $\mathcal{M}_{\text{struct}}$ to a consistency check, separate from the upstream NP-hard problem of *producing* such annotations from a logical circuit (Siraichi et al., 2018; Tan et al., 2024).

Sampling, in contrast, exhibits a sharp regime gap. In NISQ, $\mathcal{M}_{\text{struct}}$ is the conjunction of independent local constraints (one per gate), so the pass rate factors as $P(x \in \mathcal{M}_{\text{struct}}^{\text{NISQ}}) = (1 - \epsilon)^G$ in the per-gate violation probability $\epsilon$. Supervised fine-tuning on hardware-aware data can drive $\epsilon$ below $1/G$, yielding $\Theta(1)$ pass rates; this regime is empirically reached by production NISQ code-generation systems (Dupuis et al., 2024; Vishwakarma et al., 2024). In FTQC, $\mathcal{M}_{\text{struct}}$ encodes *global* combinatorial constraints such as lattice-surgery

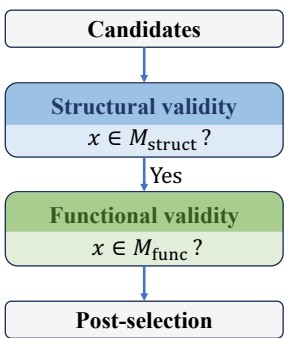

Figure 3. Two-stage post hoc filtering for quantum synthesis.

routes must not cross in spacetime, factory throughput must meet T-gate demand at every timestep, and distance budgets must sum to the target logical error rate (Litinski, 2019; Beverland et al., 2022). These constraints are not separable, and the corresponding feasibility problems inherit NP-hardness from the qubit-mapping subroutine (Siraichi et al., 2018; Botea et al., 2018), with further evidence of practical intractability from SAT-based lattice-surgery synthesis (Tan et al., 2024) and Steiner-tree formulations of multi-qubit surgery routing (Silva et al., 2024). As we show in Appendix B, this implies that no polynomial-time sampler $\pi_\theta$, including any LLM with $\mathrm{poly}(n)$ parameters and inference cost, can place more than $e^{-\Omega(n)}$ mass on $\mathcal{M}_{\mathrm{struct}}^{\mathrm{FTQC}}$ in the worst case, unless NP $\subseteq$ BPP.

The implication for the post-selection pipeline is asymmetric. In NISQ, structural filtering is essentially free: a $\Theta(1)$ pass rate combined with $\Theta(nd)$ verification yields $\Theta(nd)$ expected cost per structurally valid candidate. In FTQC, the same pipeline incurs an *additional* exponential factor before functional validity is even considered, giving expected cost $e^{\Omega(n)} \cdot \Theta(nd)$. This worst-case bound is, of course, achievable only on adversarial specifications; practical algorithms are built from regular primitives (QFT, modular arithmetic, amplitude estimation, Trotterization) whose structure admits polynomial-time schedule construction. The point of the hardness result is not that valid FTQC compilation is hopeless, but that generic probabilistic generation cannot reach it: reliability must come from algorithmic structure exploited constructively, not from scale. This motivates the verifier-centric paradigm of Section 4.

### 3.2. Functional Validity: Cost of Checking and Sampling

The $\Theta(2^n)$ verification cost in Background 2, namely the cost of comparing a circuit's induced unitary $U(x)$ against $U_{\mathrm{target}}$, is not improved by tensor-network methods: matrix product states (Vidal, 2003) require bond dimension that grows exponentially for the volume-law-entangled circuits characteristic of post-classical algorithms (Schuch et al., 2008). We therefore work with $C_{\mathrm{func}} = \Theta(2^n)$ for generic synthesis; Appendix C.1 discusses structured-target escape

routes (Clifford fragments, ZX-calculus normalization).

Verification cost is only half the problem. The conditional pass rate $P(x \in \mathcal{M}_{\mathrm{func}} \mid x \in \mathcal{M}_{\mathrm{struct}})$ also decays exponentially in $n$, by an argument whose form depends on the program representation: Lipschitz volume counting on an $O(nd)$-dimensional parameter manifold inside $SU(2^n)$ for NISQ ansatzes (Nielsen & Chuang, 2010), or direct enumeration over a combinatorial program space of size $e^{\Theta(nd \log n)}$ for FTQC instruction sequences. Both yield, under the non-pathology assumption that $\pi_\theta$ does not memorize the specific solution,

$$P_{x \sim \pi_\theta}(\|U(x) - U_{\mathrm{target}}\| \leq \varepsilon \mid x \in \mathcal{M}_{\mathrm{struct}}) \leq C_\theta \cdot e^{-\gamma n}, \quad (1)$$

where $C_\theta$ measures how far $\pi_\theta$ deviates from the uniform reference distribution on its support, bounded by $\mathrm{poly}(n)$ unless the model has memorized the specific solution, and $\gamma$ is polynomial in $d$ and logarithmic in $n$ or $1/\varepsilon$ (proofs in Appendix C.2).[1] The implication for the post-selection pipeline of Section 3.3 is direct: functional verification costs $\Theta(2^n)$ per candidate while only an $e^{-\gamma n}$ fraction of candidates passes it, so the two exponentials compound.

### 3.3. The Exponential Cost of Post-selection

Combining the structural and functional bounds of Sections 3.1 and 3.2 gives the total expected cost of the post-hoc selection pipeline (Figure 3). Writing

$$p_{\mathrm{struct}} := P(x \in \mathcal{M}_{\mathrm{struct}}),$$
$$p_{\mathrm{func}|\mathrm{struct}} := P(x \in \mathcal{M}_{\mathrm{func}} \mid x \in \mathcal{M}_{\mathrm{struct}}),$$

for the structural and conditional functional pass rates analyzed in Sections 3.1 and 3.2 respectively, and with per-candidate verification costs $C_{\mathrm{struct}} \in \Theta(nd)$ and $C_{\mathrm{func}} \in \Theta(2^n)$, the expected cost to obtain one functionally correct candidate is:

$$\mathbb{E}[C] = \frac{C_{\mathrm{struct}}}{p_{\mathrm{struct}} \cdot p_{\mathrm{func}|\mathrm{struct}}} + \frac{C_{\mathrm{func}}}{p_{\mathrm{func}|\mathrm{struct}}}. \quad (2)$$

We substitute the regime-specific bounds established earlier: $p_{\mathrm{struct}}^{\mathrm{NISQ}} = \Theta(1)$ versus $p_{\mathrm{struct}}^{\mathrm{FTQC}} \leq e^{-\beta n}$ from Proposition B.1,

---

[1] Two scope conditions are worth stating explicitly. (a) The NISQ bound requires standard regularity of the parameterization (full-rank Jacobian, bounded fiber multiplicity); pathological ansatzes in the barren-plateau regime (McClean et al., 2018) violate this, but in that regime synthesis already fails for optimization-theoretic reasons independent of our bound. (b) The bound is pointwise in $U_{\mathrm{target}}$ but informative only when $C_\theta = \mathcal{O}(\mathrm{poly}(n))$, which holds by definition of generalization for out-of-distribution targets. For in-distribution algorithmic primitives (textbook QFT, Grover, modular arithmetic), $C_\theta$ can be large enough to cancel $e^{-\gamma n}$ entirely, which is consistent with empirical success of code-completion systems on such benchmarks (Dupuis et al., 2024; Vishwakarma et al., 2024). The validity gap is therefore not about whether circuits can in principle implement $U_{\mathrm{target}}$, but about whether a sampler can find one without already having seen the answer.

where $\beta > 0$ denotes the implicit constant in its $\Omega(n)$ exponent; and $p_{\text{func}|\text{struct}} \leq C_\theta \cdot e^{-\gamma n}$ from Equation (1), with regime-specific exponents $\gamma_{\text{NISQ}} = \Omega(d\log(1/\varepsilon))$ and $\gamma_{\text{FTQC}} = \Omega(d\log n)$ from Propositions C.1 and C.2. This yields

$$\mathbb{E}[C]^{\text{NISQ}} = \mathcal{O}\Big(e^{(\ln 2 + \gamma_{\text{NISQ}})\, n}\Big), \qquad (3)$$

$$\mathbb{E}[C]^{\text{FTQC}} = \mathcal{O}\Big(e^{(\max(\ln 2,\, \beta) + \gamma_{\text{FTQC}})\, n}\Big). \qquad (4)$$

NISQ couples two exponentials (simulation cost $\ln 2$ plus functional sparsity $\gamma_{\text{NISQ}}$); FTQC couples three, with structural sparsity $\beta$ contributing alongside $\gamma_{\text{FTQC}}$ and combining with $\ln 2$ via the max — whichever of structural sampling or functional verification is more expensive dominates. Attempting to bypass these bounds via matrix product states fails for quantum-advantage regimes: useful circuits exhibit volume-law entanglement, forcing the bond dimension to grow exponentially (Vidal, 2003; Schuch et al., 2008). Specialized verifiers exist for structured subclasses (Clifford, local Hamiltonian, Clifford+T via ZX; see Appendix C.1) but not for the generic synthesis problem.

> **POSITION 2:** Probabilistic scaling cannot close the coupled exponential gap of Equation (2): structural sparsity in FTQC and functional sparsity in both regimes compound with simulation cost. We assert that validity must be enforced *constructively* during candidate generation, not *filtered* after it, shifting the verifier from an output gate to a generation invariant.

# 4. The Verifier-centric Paradigm

Current industry copilots optimize for human-centered objectives (Barke et al., 2023; Vaithilingam et al., 2022; Liang et al., 2024): variable naming, commenting, and syntactic sugar. While valuable for classical software where the human serves as the ultimate reviewer, this alignment constitutes a misallocation of resources for quantum program generation. The breakdown of human verification is not asymptotic but concrete: classical state-vector simulation, the most direct verification aid available to a human reviewer, becomes infeasible beyond roughly 40–50 qubits on current hardware (Dalzell et al., 2020; Arute et al., 2019), and recent experiments on IBM's 127-qubit Eagle experiments (Kim et al., 2023) already operate well outside this regime. At these scales, the combination of syntax-semantics gap (Section 2), the exponential sparsity of valid circuits (Section 3), and the fundamental quantum properties such as no-cloning makes manual deduction of circuit correctness intractable for any human reviewer. Human supervision becomes a bottleneck rather than a safeguard.

We posit that the objective of AI tools for rigorous science must invert. The target audience is not the human developer but the *formal verifier*. The agent functions as a high-throughput heuristic proposer for the solver, operating at machine speed and unconstrained by human verification bandwidth.

## 4.1. Constructive Verification Protocols

To surmount the exponential cost of post-selection identified in Section 3, we propose replacing blind generation with constructive verification (Figure 4). This framework adapts the constrained decoding principles (Shin et al., 2021) to domains of rigorous sciences, which have been proven effective in semantic parsing tasks like SQL generation (Scholak et al., 2021).

We formalize this process as a trajectory search through the discrete syntactic space $\mathcal{X}$. The iterative abstract quantum circuit/program design process is modeled as a Markov Decision Process $\mathcal{M} = (\mathcal{S}, \mathcal{A}, P)$ where a policy is a conditional distribution $\pi_\theta(a_t|s_t)$ that samples an action $a_t \in \mathcal{A}$ at time step $t$ given the current state $s_t \in \mathcal{S}$ and the environment dynamics are given by a transition kernel $P(s_{t+1}|s_t, a_t)$ (Puterman, 1994; Sutton & Barto, 2018).

- **State space $\mathcal{S}$.** A state $s_t$ is the agent's current design snapshot, denoted as

$$s_t := \big(x_t,\ \phi_t,\ \eta_t,\ \kappa_t\big),$$

where $x_t \in \mathcal{X}$ is the current abstract program/circuit is in a fixed intermediate representation (IR), $\phi_t$ has current formal parameters (e.g., input size $n$, error budgets, success probability targets), $\eta_t$ stores the latest evaluation outputs/metrics (depth, $T$-count, ancilla count, asymptotic scaling estimates), and $\kappa_t$ has any auxiliary bookkeeping needed to make the process Markov (e.g., tool call traces, module-commitment flags).

- **Action space $\mathcal{A}$.** Each action corresponds to invoking a *tool* from a fixed library with parameters, e.g.

$$a_t \in \{\text{EDIT}[\rho, \alpha],\ \text{EVAL},\ \text{FINISH}\},$$

where $\text{EDIT}[\rho, \alpha]$ applies a property-preserving rewrite schema $\rho$ with arguments $\alpha$ (chosen from a verified transformation library), see Figure 5c. EVAL re-checks correctness and evaluates metrics, and FINISH terminates and returns the final template and summary.

- **Dynamics $P(s_{t+1}|s_t, a_t)$.** The transition kernel is induced by the tool execution and its guardrails:

$$s_{t+1} \sim P(\cdot|s_t, a_t) \quad \Leftrightarrow \quad s_{t+1} = \text{TOOLSTEP}(s_t, a_t).$$

Concretely, for an edit action $a_t = \text{EDIT}[\rho, \alpha]$ for some $\rho$ and $\alpha$, $\text{TOOLSTEP}(s_t, a_t) = \big(x'_t, \phi_t, \eta_t, \kappa'_t\big)$ if the rewrite is admitted and applied, and $\text{TOOLSTEP}(s_t, a_t) = \big(x_t, \phi_t, \eta_t, \kappa''_t\big)$ if the action is blocked/rejected (no-op).

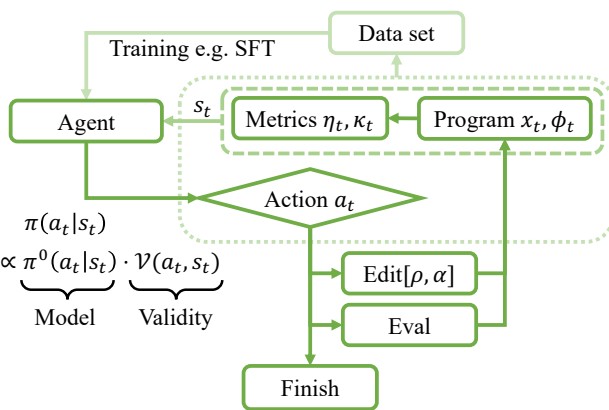

*Figure 4.* **Control flow of the verification protocol.** This can be considered as an expanded view of Figure 2 where the decision process of the verifier-centric agent is expanded in greater detail.

- **Validity as a state-dependent admissible action set.** The semantic constraints discussed in Sections 2 and 3 imply that, for each state $s_t$, there is an admissible action subset $\mathcal{A}_{\text{valid}}(s_t) \subseteq \mathcal{A}$ to ensure that $x_{t+1} \in \mathcal{M}_{\text{func}}$. We implement this through a *validity filter* (a.k.a. shield / mask)

$$\mathcal{V}(a, s) \in \{0, 1\}, \quad \mathcal{A}_{\text{valid}}(s) := \{a \in \mathcal{A} : \mathcal{V}(a, s) = 1\}.$$

The filter may be composed of multiple components and dependent on the action $a_t$ taken. For instance, if $a_t = \text{EDIT}(\text{Parallelize}, (g_1, g_2))$ for some chosen pair of gates $g_1$, $g_2$ in the circuit $x_t$, namely if the action is to try parallelizing $g_1$ and $g_2$, the validity filter needs to check 1) that the two gates operate on distinct sets of qubits ("no-overlap") and 2) there is no obstruction for the two gates to be moved to the same layer ("no-obstacle"). In this case the validity filter takes the form of

$$\mathcal{V}(a, s) = \mathcal{V}_{\text{no-overlap}}(a, s) \cdot \mathcal{V}_{\text{no-obstacle}}(a, s), \quad (5)$$

Here we expect each component of the validity filter to run in $\text{poly}(n)$ time. The components of the validity filters in (5) clearly run in polynomial time.

- **Constrained policy.** Let $\pi_\theta^{(0)}(a|s)$ be the unconstrained LLM-induced proposal over tool calls. The constructive-verification policy is proportional to the unconstrained proposal multiplied by the validity filter (up to normalization):

$$\pi_\theta(a|s) \propto \pi_\theta^{(0)}(a|s) \cdot \mathcal{V}(a, s). \quad (6)$$

The product form of the constrained policy in (6) is reminiscent of the standard invalid-action masking construction in reinforcement learning (Huang & Ontañón, 2022; Hou et al., 2023).

## 4.2. Multiple Levels of Abstraction

Even though the semantics of an arbitrary $n$-qubit circuit quickly becomes intractable to visually or exhaustively validate as $n$ grows, practical quantum programs are rarely arbitrary: they are typically assembled from a small vocabulary of algorithmic primitives with a strong subroutine containment structure. As an example, the circuit for Shor's algorithm (Nielsen & Chuang, 2010, Chapter 5) is built from quantum fourier transform (QFT) and modular exponentiation blocks. Modular exponentiation then decomposes into modular multiplication and modular addition, each with well-understood reversible implementations and local invariants (e.g., ancilla management and uncomputation patterns) that can be verified and constrained module-by-module rather than by reasoning about an entire unitary monolithically. This multi-level containment is the key escape hatch from the brute-force complexity discussed in Section 3: instead of handling a large quantum circuit as a monolith, we validate and constrain generation at the granularity of primitives (QFT, modular addition/multiplication/exponentiation, etc.), and then compose those certificates—exactly the kind of hierarchical abstraction that a verifier-centric agent must exploit.

**Level 1: module based circuit design.** At the top level, the agent manipulates a parametric quantum program with $n$ treated as a formal parameter (alongside other algorithmic parameters), and subroutines are invoked as black boxes. Verification at this level is therefore structural: the agent checks that the correct high-level subroutine calls appear in the correct order and with consistent wiring of symbolic registers, without expanding the subroutines into gates. For the Cuccaro ripple-carry adder (Cuccaro et al., 2004), the $n$-bit in-place addition can be written as a short program over two 3-bit modules, MAJ and UMA, applied in forward/backward sweeps (Figure 5a). At this abstraction, the agent reasons about containment where the adder is composed of MAJ/UMA modules.

**Level 2: decomposition of individual modules.** Level 2 is the first level where the focus shifts to circuits on $O(1)$ qubits: each black-box module is compiled into a concrete gate-level implementation. For the adder, this means choosing decompositions of the 3-qubit blocks MAJ and UMA into elementary gates (e.g., CNOT/Toffoli or backend-native one- or two-qubit gates), exactly as illustrated in Figure 5b. This is the regime where typical quantum gate transpilers operate: correctness obligations are localized to a small footprint, and the agent can use standard synthesis/transpilation toolchains (Javadi-Abhari et al., 2024) to map each module to a target gate set and connectivity.

**Level 3: elementary-operation optimization (property-preserving rewrites).** At Level 3, the program is treated as a monolithic sequence of elementary operations obtained

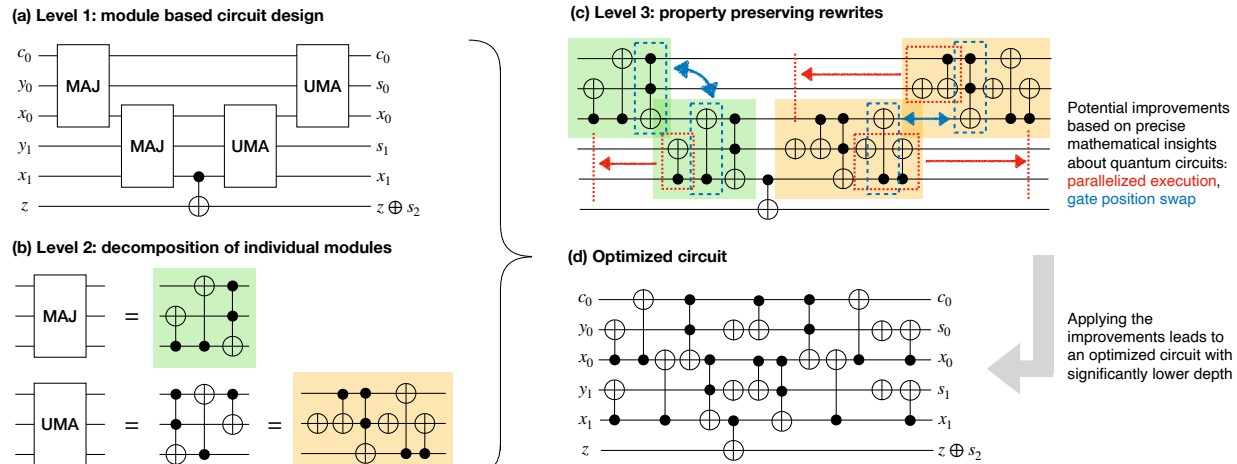

*Figure 5.* Illustration of the different levels of abstraction using the Cuccaro Adder (Cuccaro et al., 2004) as an example. **(a)** Level 1 in the hierarchy describes the abstract circuit design based on modules MAJ and UMA. Here a concrete example with $n = 2$ is shown. **(b)** Level 2 expands each module into more elementary gate operations. **(c)** Level 3 identifies opportunities for improvements and acts on those opportunities with property-preserving rewrites $\rho$. Here $\rho \in \{\text{Parallelize}, \text{Swap}\}$ and the parameter set $\alpha$ passed depends on which $\rho$. **(d)** Applying the improvements (parallelized execution of gates and gate position swap due to commutativity) yields an optimized circuit with lower depth. We also provide a detailed example of how such property perserving rewrites can be applied to the Cuccaro ripple-carry adder in Appendix Section D.

by inlining the Level 2 decompositions into the Level 1 program. The agent then searches for depth/gate-count improvements via property-preserving rewrites and local commutation or parallelization opportunities (Figure 5c): e.g., swapping commuting gates, canceling inverse pairs, sliding CNOTs across Toffolis when allowed, and parallelizing disjoint operations. Crucially, Level 3 remains aware of higher-level choices: selecting a different Level 2 decomposition (or changing the Level 1 module schedule) can expose or eliminate rewrite opportunities, so optimization is inherently cross-level even though the rewrite rules act on elementary gates.

To illustrate how the control framework outlined in Section 4.1 can be applied across multiple layers of abstraction (Figure 5), a detailed worked example is shown in Section D in the Appendix. Here the goal is to demonstrate the form of the program $x_t$ and how property preserving rewrites can help the agent work through different layers of abstraction.

### 4.3. The Data Imperative

Our earlier work (Song et al., 2026) argued for verified training data as an architectural primitive rather than a curation afterthought. The present work supplies its complexity foundation. Standard code-generation pipelines rely on syntactic supervision, which suffices for classical software but collapses in scientific domains where syntactic fluency is decoupled from physical semantics (Section 2). A model trained on such data mimics how humans write code while ignoring the underlying constraints. We therefore advocate

trace-based datasets, analogous to process reward models in mathematical reasoning (Lightman et al., 2023). Data for scientific agents must contrast optimal solutions against valid but sub-optimal candidates, encoding the optimization manifold rather than a binary correctness function.

Narrowing the syntax–semantics gap forces the agent to internalize hard domain constraints, not just surface syntax. This is the broader AI4Science mandate. Scientific data is scarce, so inductive bias must be supplied a priori. Otherwise, scaling only produces hallucinations at a larger scale.

> **POSITION 3:** AI4Science must transition from open-loop generation to closed-loop constructive verification. We contend that integrating constraints across multiple levels of abstraction is a viable path to scale modular quantum synthesis beyond the limits of human cognition.

## 5. Alternative Views

This paper proposes to shift from human-centric copilots to verifier-centric agents, which challenges the generative AI (GenAI) orthodoxy. Below, we address three common counter-arguments. We contrast GenAI wisdom with the strict requirements of quantum program generation.

**View 1: Scaling maximalist.** In the machine learning community, we often hear that the model with enough pa-

rameters will have an "Aha" moment to understand the logic of events or physics rules (quantum circuits) purely (Wei et al., 2022). Many researchers argue that specialized verification is unnecessary. They posit that with sufficient parameter scaling, models also will eventually "grok" the logic of quantum circuits purely by training on vast corpora of QASM and Python code.

**Rebuttal: The inverse scaling of quantum correctness.** This extrapolation conflates natural language fluency with physical validity. Large models often exhibit inverse scaling on tasks with strict logical constraints (McKenzie et al., 2023; Zhou et al., 2024), becoming more confident in their hallucinations (Dziri et al., 2023). A larger model may emit syntactically perfect QASM yet fail to implement the target unitary due to subtle phase errors or invalid gate decompositions (see Section 2 and Figure 2). Scaling amplifies stylistic mimicry while remaining blind to Hilbert-space constraints, and the same pathology surfaces at inference, where scaling test-time compute pushes models into unfocused exploration (Gema et al., 2025). The admissible action set $\mathcal{A}_{\mathrm{valid}}(s)$ in Section 4 rules out both failure modes by construction.

**View 2: Developer experience advocate.** A common objection from the software engineering and human–AI interaction communities holds that coding assistants must remain legible and interactive to the human developer, even if final correctness adjudication is delegated to a solver. Optimizing purely for a formal verifier risks producing opaque artifacts that erode developer trust.

**Rebuttal: Legibility lives in the trace, not the artifact.** We do not eliminate the human; we relocate their role from correctness adjudicator (already intractable beyond $\sim$50 qubits, Section 4) to specification author and trace auditor. The trace-based datasets advocated in Section 4.3 record *why* each rewrite was applied and *which alternatives were rejected*, providing a richer interpretability surface than the final circuit. The human reads the reasoning, not the unitary — strictly more informative than syntactic copilots, which provide neither correctness guarantees nor process-level traces (Lightman et al., 2023).

**View 3: Computational pragmatist.** Critics might argue that running formal verifiers during generation is computationally too expensive, preferring the less costly inference of standard deep learning models followed by post-hoc filtering.

**Rebuttal: The economics of quantum discovery.** We argue that the "Generate then Filter" paradigm is mathematically ruinous for quantum search spaces, or generally in domains where valid solutions are rare (rejection sampling is notoriously inefficient). AlphaCode required over one million samples to solve hard competitive problems

(Li et al., 2022). Bio-inspired metaheuristics (Song et al., 2024) hit the same exponential wall. In quantum computing, this inefficiency is exponentially worse. As shown in Section 3, the valid subspace decays as $e^{-\gamma n}$. Brute-force sampling is not just expensive, it is intractable compared to verification-aware generation.

**View 4: Quantum verification is a solved problem.** Mahadev (Mahadev, 2018) gives a classical verifier for arbitrary BQP computations under post-quantum cryptographic assumptions, with follow-up extending the protocol to blind, succinct, and non-interactive settings (Fitzsimons & Kashefi, 2017; Reichardt et al., 2013; Bartusek et al., 2022). If verification is already solved, why claim it as the bottleneck?

**Rebuttal: Execution verification is not functional verification.** That line verifies a prover *honestly executed a specified circuit* $C$. Our concern is the orthogonal problem: given a functional specification $U_{\mathrm{target}}$, decide whether a candidate $C$ implements it. Mahadev's protocol takes $C$ as input and is silent on whether $C$ was the right program in the first place. This is the gap between $\mathcal{M}_{\mathrm{struct}}$ and $\mathcal{M}_{\mathrm{func}}$ in Section 4; the QMA-completeness of non-identity testing (Watrous, 2009) applies directly to the latter and is not lifted by cryptographic execution verification.

## 6. Conclusion

We have argued that probabilistic scaling alone is unlikely to close the validity gap in quantum program generation. Training on unverified code introduces a structural bias against physical validity, and the exponential sparsity of functionally correct circuits makes post-hoc filtering intractable for generic synthesis. Larger models trained this way fit a biased distribution more faithfully, rather than converging on the manifold of valid programs.

We therefore advocate a shift from generate-then-filter pipelines to verifier-centric generation. Solving arbitrary circuits remains out of reach, but the hierarchical, module-based approach of Section 4 is tractable for the broad class of algorithms built from well-understood primitives, by enforcing topological and symbolic constraints during decoding. Trace-based datasets, which record why a rewrite was chosen and why alternatives were rejected, offer a natural training signal for this regime.

> **In domains with hard physical and mathematical constraints, generative models should be coupled to constructive verification rather than relying on scale to recover validity after the fact.**

## Impact Statement

Scaling on corpora will not close the gap between fluency and correctness when validity is governed by mathematical structure rather than statistical regularity. Larger models trained on unverified quantum code yield more confident hallucinations, not useful software. The same failure recurs wherever valid outputs occupy a vanishing fraction of a high-dimensional space, including protein design, theorem proving, and chip layout. Progress there will come from smaller systems coupled to formal checkers, not from larger imitators. A verifier-centric paradigm relocates trust onto the verifiers themselves. A subtle solver bug can silently certify invalid designs at scale, so auditing the verifier stack must sit alongside model training as a research priority. Constraint-aware generation makes outputs easier to audit, which reduces rather than increases risk.

## Acknowledgements

Many thanks to the Imperial College Centre for Cryptocurrency Research and Engineering for research support. Thanks to Runzhou Tao (University of Maryland) for early discussion. Junhao Song is a first-year PhD student fully funded by the Hitachi-Imperial Centre for Decarbonisation and Natural Climate Solutions, a collaboration between Hitachi Ltd, Hitachi Europe and Imperial College London.

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

# Appendices

## A. The Fallacy of Local Modularity.

Below we use an example to illustrate a fundamental difference between quantum and classical programming (also see Figure 2 for a summary).

**The classical regime (local modularity):** Consider the classical bit program: `x = y; temp = x;` where `x`, `y`, and `temp` are bits. Classically, both lines are syntactically and semantically acceptable under routine inspection: the assignment `temp = x` may be unused without invalidating the program (at most producing a compiler warning). More broadly, classical semantics tolerates *erasure*: overwriting `x` discards the old value of `x` with no global obligation to retain invertibility.

**The quantum reality (global interference):** This analogy collapses in quantum computing because valid programs must extend to *reversible* (unitary) transformations. Overwriting a register (e.g., `x := y`) is not a bijection unless the overwritten information is preserved elsewhere (Bennett, 1973) and any temporary workspace is later *uncomputed* to decouple it from the computational register. A syntactically correct circuit can therefore be globally invalid in the sense of leaving residual entanglement that destroys later interference.

- *Failure Mode (discarding a "harmless" temp):* A common reversible embedding of the classical intent is to compute into ancillas and then uncompute intermediate work. Suppose an LLM produces the following circuit fragment intended to mirror the classical program:

```
CNOT(y, x)    # ``x = y''
CNOT(x, temp) # ``temp = x''
CNOT(y, x)    # uncompute ``x = y''
```

  The last line correctly restores `x` to its original value, but the fragment *ignores* the fact that `temp` now carries information about `y`. If `y` is in a superposition $\frac{1}{\sqrt{2}}(|0\rangle + |1\rangle)$ and `temp` starts at $|0\rangle$, the map produces the entangled state

$$\frac{1}{\sqrt{2}}(|0\rangle_y |0\rangle_{\text{temp}} + |1\rangle_y |1\rangle_{\text{temp}}),$$

  so `temp` is **not** a benign leftover variable: it is a *which-branch record*. Any subsequent algorithm that relies on interference in the `y` register will generally fail because coherence has leaked into `temp`.
- *Consequence (non-local semantic constraints):* In classical code, leaving an unused `temp` is harmless; in quantum circuits, leaving a "temporary" register entangled with the computational state changes the reduced state of the register and can destroy the interference pattern required for the computation. This is a *global* semantic failure: no local syntax parser can detect it, and it is not repaired by superficial fluency.
- *Implication for LLMs (why scale is not enough):* Next-token training optimizes for *syntactic likelihood*, not for satisfaction of hard mathematical invariants such as reversibility/unitarity (and, on hardware, topology and scheduling constraints). The toy example already exhibits the core mismatch: correctness depends on a global bijection and explicit uncomputation, not on locally plausible lines. Scaling up LLMs can improve stylistic fidelity, but it does not by itself enforce the required algebraic constraints; reliable generation must instead couple decoding to explicit verifiers/solvers and uncomputation-aware compilation, i.e., a *verification-centric* loop rather than "generate-then-hope".
- *Efficient identification of the failure mode:* Unlike general quantum programs, which costs exponential time to simulate, failure modes such as this example can be identified without exponential-time simulation. Concretely, one can track the Pauli stabilizer $Z$ acting on qubit $y$ throughout the circuit, since the circuit consists entirely of Clifford gates and is therefore efficiently simulable via the Gottesman–Knill theorem. Ordering the qubits as $(y, x, \text{temp})$, the stabilizer evolution is:

$$ZII \xrightarrow{\text{CNOT}(y,x)} ZZI \xrightarrow{\text{CNOT}(x,\text{temp})} ZZZ \xrightarrow{\text{CNOT}(y,x)} ZIZ.$$

If the intended computation had succeeded, the temporary register would be cleanly isolated at the end, and we would expect the final operator to be $IIZ$, indicating that only `temp` carries the residual information. Instead, the actual final operator is $ZIZ$, which shows that the output still has support on both $y$ and `temp`.

## B. Structural Sampling Hardness in NISQ and FTQC

This appendix expands the structural-sampling claim of Section 3.1. We first formalize the verification cost and pass-rate decomposition for both regimes, then prove the worst-case sampling cost for FTQC.

### B.1. Verification Cost Decomposition

For a candidate circuit description $x$ on $n$ logical qubits, depth $d$, and $G = \Theta(nd)$ operations, the cost of deciding $x \in \mathcal{M}_{\text{struct}}$ decomposes additively over the local constraints in $x$:

$$C_{\text{struct}}(x) = C_{\text{parse}}(x) + \sum_{k=1}^{K} C_k(x), \tag{7}$$

where $C_{\text{parse}}(x) = \Theta(|x|) = \Theta(nd)$ is the cost of reading the circuit representation, often described in the language of some intermediate representation (IR), and $C_k(x)$ is the cost of the $k$-th constraint check.

**NISQ checks.** The relevant constraint classes are:

*Table 1.* NISQ structural checks. $G$ is total gate count, $G_2 \leq G$ is the number of two-qubit gates.

| Check | Per-operation cost | Total |
|---|---|---|
| Gate-set membership | $\mathcal{O}(1)$ table lookup | $\Theta(G)$ |
| Coupling-graph adjacency | $\mathcal{O}(1)$ hash lookup | $\Theta(G_2)$ |
| Wire well-formedness | $\mathcal{O}(1)$ | $\Theta(G)$ |
| Coherence-time budget (optional) | $\mathcal{O}(1)$ | $\Theta(G)$ |

*Table 2.* FTQC structural checks. $F$ is the number of magic-state factories, $d$ is circuit depth.

| Check | Per-op or per-step cost | Total |
|---|---|---|
| Patch-layout adjacency | $\mathcal{O}(1)$ | $\Theta(G)$ |
| Boundary-type matching | $\mathcal{O}(1)$ | $\Theta(G)$ |
| Code-distance budget per op | $\mathcal{O}(1)$ arithmetic | $\Theta(G)$ |
| T-throughput per timestep | $\mathcal{O}(F)$ | $\Theta(Fd)$ |
| Routing-channel availability | $\mathcal{O}(1)$ per surgery | $\Theta(G)$ |
| Pauli-frame consistency | $\mathcal{O}(1)$ per op | $\Theta(G)$ |

where $F$ is the number of magic-state factories. Under the standard scaling $F = \mathcal{O}(n)$ (Beverland et al., 2022), the total is $\Theta(nd + Fd) = \Theta(nd)$. Crucially, all checks are local in $x$: each references a constant number of operations or a single timestep slice. None require simulating the circuit or computing information not already present in $x$.

We treat the layout, schedule, and Pauli-frame annotations as part of the candidate description $x$, separating verification from the upstream synthesis problem of *producing* such annotations from an unannotated logical circuit. The latter is NP-hard in both regimes (Siraichi et al., 2018; Ito et al., 2023; Tan et al., 2024) and is upstream of $\mathcal{M}_{\text{struct}}$ membership.

### B.2. NISQ Pass Rate

In NISQ, $\mathcal{M}_{\text{struct}}$ is a conjunction of independent local constraints, one per gate (more precisely, per two-qubit gate for the coupling-graph constraint, and per gate for the gate-set and wire constraints). If $\epsilon$ denotes the per-gate violation probability of a model fine-tuned on coupling-graph-aware data, the structural pass rate is

$$P(x \in \mathcal{M}_{\text{struct}}^{\text{NISQ}}) = \prod_{g \in x}(1 - \epsilon_g) \geq (1 - \epsilon)^G, \tag{8}$$

where $\epsilon = \max_g \epsilon_g$. For $\epsilon = o(1/G)$, this converges to $\Theta(1)$ as $G \to \infty$. Empirically, NISQ-targeted code-generation systems such as Qiskit Code Assistant achieve near-unity structural pass rates after task-specific fine-tuning (Dupuis et al., 2024; Vishwakarma et al., 2024), consistent with the theoretical scaling.

### B.3. FTQC Sampling Hardness

The FTQC structural constraints are not separable per-gate: a circuit may be locally legal at every operation yet globally infeasible because, for instance, three lattice surgeries demand the same routing channel at the same timestep, or the cumulative T-gate demand exceeds factory throughput at some peak load. This non-separability is what gives rise to the NP-hardness of the underlying feasibility problems.

**Proposition B.1** (Structural sampling hardness in FTQC). *There exists a polynomial-time-constructible family of FTQC compilation specifications $\{\Sigma_n\}_{n\geq 1}$, on $n$ logical qubits and depth $d_n = \text{poly}(n)$, with the following property. Let $\pi_\theta$ be any distribution over candidate circuit descriptions that can be sampled in time $\text{poly}(n)$. Then either*

$$P_{x\sim\pi_\theta}\left(x \in \mathcal{M}_{struct}^{FTQC}(\Sigma_n)\right) \leq e^{-\Omega(n)} \qquad \text{for sufficiently large } n, \tag{9}$$

*or* $\mathsf{NP} \subseteq \mathsf{BPP}$.

*Proof.* We reduce 3-SAT to the FTQC structural-feasibility problem and combine with standard BPP-amplification.

**Reduction.** Let $\phi$ be a 3-SAT instance on $m$ variables and $\text{poly}(m)$ clauses. Qubit allocation: given a logical quantum circuit and a hardware coupling graph, decide whether a connectivity-respecting allocation exists, which is NP-complete via a polynomial-time reduction from subgraph isomorphism (Siraichi et al., 2018; Botea et al., 2018). Composing this with the standard linear-blowup reduction from 3-SAT through constraint-graph encoding yields, in $\text{poly}(m)$ time, an FTQC compilation specification $\Sigma(\phi)$ on $n = \Theta(m)$ logical qubits and depth $d = \text{poly}(m)$, such that the set of descriptions $x$ satisfying $x \in \mathcal{M}_{struct}^{FTQC}(\Sigma(\phi))$ is in $\text{poly}(m)$-time computable bijection with the set of satisfying assignments of $\phi$. The surface-code lattice-surgery substrate inherits this hardness because structurally valid descriptions must include a patch-to-tile assignment respecting the lattice's adjacency structure (Litinski, 2019; Tan et al., 2024); the SAT encoding of Tan et al. (2024) establishes that this subproblem is at least as hard as general Boolean satisfaction in practice.

**Amplification.** Suppose for contradiction that there exists a $\text{poly}(n)$-time-samplable $\pi_\theta$ satisfying $P(x \in \mathcal{M}_{struct}^{FTQC}(\Sigma(\phi))) \geq 2^{-cn}$ for some constant $c$ and all sufficiently large $n$. Then for $\phi$ on $m$ variables, repeated independent sampling from $\pi_\theta$ followed by $\Theta(nd) = \text{poly}(m)$-time structural verification (Section B.1) yields a satisfying assignment of $\phi$ in expected $2^{cn} = 2^{\Theta(m)}$ trials. For $c < 1$, this is $o(2^m)$ expected work, contradicting the Exponential Time Hypothesis; for the unconditional statement that $\mathsf{NP} \not\subseteq \mathsf{BPP}$, take $c$ any constant and amplify to constant success probability via $\text{poly}(n)$ independent repetitions, which contradicts $\mathsf{NP} \not\subseteq \mathsf{BPP}$. $\square$

**Remarks.**

**(i) The proposition is a worst-case statement over compilation specifications.** Practical quantum programs have regular structure (repeated QFT blocks, regular Trotter steps, structured amplitude estimation) that admits polynomial-time schedule construction (Litinski, 2019; Beverland et al., 2022). The verifier-centric paradigm of Section 4 exploits this structure constructively.

**(ii) The hardness is structural, not quantum.** The same argument applies to any NP-hard discrete satisfaction problem embedded into a sampling target. What is quantum-specific is the natural emergence of such constraints from physical considerations (lattice surgery, magic-state distillation, error-correction thresholds), making FTQC a setting where the structural barrier is unavoidable rather than artificial.

**(iii) The exponent $\Omega(n)$ in Eq. (9) depends on the qubit-blowup of the underlying reduction.** The construction sketched in the proof uses qubit allocation NP-hardness (Siraichi et al., 2018; Botea et al., 2018) composed with the surface-code substrate; tighter encodings (e.g., direct SAT encodings of lattice-surgery scheduling (Tan et al., 2024)) may yield sharper constants. The precise exponent is not load-bearing for the argument: any exponential decay suffices to make worst-case FTQC post-selection exponentially worse than the NISQ case (Section 3.3).

## C. Functional Validity Analysis

This appendix expands Section 3.2. Appendix C.1 discusses the verification cost in detail, including specialized cases where the $\Theta(2^n)$ bound can be beaten. Appendix C.2 states and proves the formal sparsity bound underlying Equation (1).

## C.1. Verification Cost $C_{\text{func}}$

**Exact verification.** Given a candidate circuit $x$ on $n$ qubits at depth $d$ and a target unitary $U_{\text{target}} \in SU(2^n)$, the most direct verification scheme is exact state-vector simulation: maintain a length-$2^n$ complex vector, apply each of the $\Theta(nd)$ gates as a sparse linear operator, and compare the final state against $U_{\text{target}}|\psi\rangle$ for a basis of input states $|\psi\rangle$. The per-gate cost is $\Theta(2^n)$ for generic two-qubit gates, giving total cost $\Theta(nd \cdot 2^n)$. Memory is $\Theta(2^n)$. Both scale exponentially in $n$, becoming infeasible beyond approximately 40–50 qubits on current classical hardware (Dalzell et al., 2020; Arute et al., 2019).

**Approximate decision is also hard.** A weaker verification target: merely deciding whether $\|U(x) - U_{\text{target}}\| \leq \varepsilon$ versus $\|U(x) - U_{\text{target}}\| \geq \varepsilon'$ for some gap $\varepsilon' > \varepsilon$, does not escape the exponential barrier. The complementary problem (non-identity check) is QMA-complete (Watrous, 2009): a polynomial-time classical algorithm deciding non-identity for arbitrary polynomial-depth circuits would imply QMA $\subseteq$ P. Even with a quantum verifier and polynomial quantum advice, the problem is not known to be in P.

**Tensor networks fail in the relevant regime.** Matrix product state (MPS) and more general tensor-network simulators reduce verification cost to $\text{poly}(n, \chi)$, where $\chi$ is the bond dimension required to faithfully represent the intermediate states. For circuits whose intermediate states have entanglement entropy scaling as the boundary of a subregion (area law), $\chi$ is bounded by a constant or polynomial in $n$, and verification is efficient. However, useful post-classical quantum algorithms generate volume-law entanglement: the entropy of a contiguous subregion of $k$ qubits scales as $\Theta(k)$ rather than $\Theta(\partial k)$ (Schuch et al., 2008). Volume-law entanglement forces $\chi = e^{\Omega(n)}$ (Vidal, 2003), returning the simulation cost to $\Theta(2^n)$. This is not a limitation of MPS specifically: any tensor network whose contraction cost is polynomial in $\chi$ inherits the same exponential blowup for volume-law states.

**Specialized cases that escape the exponential.** For completeness, we note three regimes where $C_{\text{func}}$ is polynomial:

- *Clifford fragments.* Circuits composed entirely of Clifford gates (CNOT, H, S) admit polynomial-time stabilizer simulation by the Gottesman–Knill theorem. Verification of Clifford subcircuits against Clifford targets is $\text{poly}(n, d)$. This is exploited in our Appendix Example 1 to identify the uncomputation failure mode without exponential simulation.

- *Structured Hamiltonian targets.* For Hamiltonian simulation of local Hamiltonians on geometrically structured lattices, the target $U_{\text{target}} = e^{-iHt}$ has known polynomial-time verifiers via local observables, provided the simulation time $t$ is short enough that the Lieb–Robinson cone remains local.

- *ZX-calculus normalization.* For Clifford+T circuits, the ZX-calculus admits polynomial-time normalization to canonical form, enabling functional equivalence checking in $\text{poly}(n, d)$ time for the Clifford sector and exponential time only for the T-count.

These regimes do not invalidate the $\Theta(2^n)$ bound: they describe *structured* subspaces of $\mathcal{M}_{\text{func}}$, and the generic synthesis problem this paper addresses — arbitrary $U_{\text{target}} \in SU(2^n)$ without prior structural knowledge — remains exponential. Indeed, the existence of polynomial-time verifiers for structured cases is consistent with our broader thesis (Section 4): exploitable structure must be encoded *constructively* into generation, not discovered post hoc.

## C.2. The Semantic Entropy Gap

We formalize the bound stated in Equation (1) in two settings that together cover the practical program generation pipeline: the continuous parameter manifold (Appendix C.2.1, natural for variational NISQ algorithms with parameterized rotation gates) and the discrete program description (Appendix C.2.2, natural for FTQC and for any high-level program representation operating over a finite instruction set). The two settings differ in mathematical machinery: Lipschitz volume counting vs. combinatorial enumeration, but yield the same qualitative bound.

**Common setup.** In both settings, $U_{\text{target}} \in SU(2^n)$ is the target unitary, $\pi_\theta$ is the model's distribution conditioned on $\mathcal{M}_{\text{struct}}$, and $f$ denotes the representation-to-unitary map. The bound is stated for a pointwise $U_{\text{target}}$ (no Haar or random-distribution assumption). The non-pathology constant $C_\theta$ measures how far $\pi_\theta$ deviates from the uniform reference distribution on its support; the assumption $C_\theta = \mathcal{O}(\text{poly}(n))$ expresses that the model has not memorized the specific solution.

### C.2.1. NISQ ANALYSIS: CONTINUOUS PARAMETER MANIFOLD

**Setup.** Fix circuit depth $d$ and let $\mathcal{A}_d^{\text{NISQ}}$ denote the parameter manifold of structurally valid $n$-qubit ansatzes at depth $d$ with continuous rotation parameters. Each parameterized gate contributes $\mathcal{O}(1)$ continuous parameters and the ansatz contains $\Theta(nd)$ gates, so $\dim_{\mathbb{R}} \mathcal{A}_d^{\text{NISQ}} = D \leq c_0 nd$ for a constant $c_0$ independent of $n$ (Nielsen & Chuang, 2010). Compilation defines a smooth map $f^{\text{NISQ}} : \mathcal{A}_d^{\text{NISQ}} \to SU(2^n)$. We bound the conditional pass rate by the Radon–Nikodym derivative against the uniform base measure $\mu$:

$$P_{x \sim \pi_\theta}(\|U(x) - U_{\text{target}}\| \leq \varepsilon) \leq C_\theta \cdot \frac{\mu\big((f^{\text{NISQ}})^{-1}(B_\varepsilon(U_{\text{target}}))\big)}{\mu(\mathcal{A}_d^{\text{NISQ}})}, \tag{10}$$

where $C_\theta := \|d\pi_\theta / d\mu\|_\infty$.

**Proposition C.1** (Semantic entropy gap, NISQ). *Under the setup above, for any $U_{target} \in SU(2^n)$ and any $\varepsilon \in (0, 1)$,*

$$P_{x \sim \pi_\theta}(\|U(x) - U_{target}\| \leq \varepsilon \mid x \in \mathcal{M}_{struct}) \leq C_\theta \cdot e^{-\gamma_{NISQ} n}, \tag{11}$$

*with $\gamma_{NISQ} = \Omega(d \log(1/\varepsilon))$.*

*Proof.* We assume two regularity conditions on $f^{\text{NISQ}}$: **(R1)** *generic full rank*, i.e., $J_{f^{\text{NISQ}}}(x) := \sqrt{\det(df^T df)} \geq j_0 > 0$ $\mu$-almost everywhere; and **(R2)** *bounded fiber multiplicity*, $\#((f^{\text{NISQ}})^{-1}(y)) \leq N_{\text{fiber}}$ for $\mathcal{H}^D$-a.e. $y$ in the image. For ansatzes with bounded, non-redundant generators, $j_0 = \Omega(1)$ and $N_{\text{fiber}} \leq 2^{O(d)}$ by a Bezout-type bound on the algebraic degree of $f^{\text{NISQ}}$ as a trigonometric map. Pathological cases (e.g., barren-plateau ansatzes with $j_0 = e^{-\Omega(n)}$ (McClean et al., 2018)) violate (R1) and are addressed in Remark (v).

If $U_{\text{target}} \notin \overline{f^{\text{NISQ}}(\mathcal{A}_d^{\text{NISQ}})}$, the preimage is empty and the bound is trivial. Otherwise, under (R1) the image is a $D$-dimensional immersed submanifold of $SU(2^n)$ with $D = \Theta(nd) \ll 4^n - 1$. By Federer's area formula (Federer, 1969), for any Borel $A \subseteq SU(2^n)$:

$$\int_{(f^{\text{NISQ}})^{-1}(A)} J_{f^{\text{NISQ}}}(x) \, d\text{vol}(x) \;=\; \int_{A \cap f^{\text{NISQ}}(\mathcal{A}_d^{\text{NISQ}})} \#\big((f^{\text{NISQ}})^{-1}(y)\big) \, d\mathcal{H}^D(y), \tag{12}$$

where vol is Lebesgue measure on a product chart $\mathcal{A}_d^{\text{NISQ}} \subseteq [0, 2\pi]^G$ with $G = D = c_0 nd$, and $V_0 := \text{vol}(\mathcal{A}_d^{\text{NISQ}}) = (2\pi)^G$. Setting $A = B_\varepsilon(U_{\text{target}})$, applying (R1) on the LHS and (R2) together with the standard submanifold volume bound $\mathcal{H}^D(B_\varepsilon \cap M) \leq \omega_D \varepsilon^D$ for $M$ of bounded extrinsic curvature (Federer, 1969, §3.2.39) on the RHS:

$$j_0 \cdot \text{vol}\big((f^{\text{NISQ}})^{-1}(B_\varepsilon)\big) \;\leq\; N_{\text{fiber}} \cdot \omega_D \varepsilon^D, \tag{13}$$

where $\omega_D = \pi^{D/2}/\Gamma(D/2 + 1) \leq O(1)$. Converting to the probability measure $\mu = \text{vol}/V_0$:

$$\mu\big((f^{\text{NISQ}})^{-1}(B_\varepsilon(U_{\text{target}}))\big) \;\leq\; \frac{N_{\text{fiber}}}{j_0} \cdot \left(\frac{\omega_D^{1/D} \varepsilon}{2\pi}\right)^D. \tag{14}$$

Substituting into Equation (10) and taking logs:

$$-\log P \;\geq\; c_0 nd \log \frac{2\pi}{\omega_D^{1/D} \varepsilon} - \log \frac{N_{\text{fiber}}}{j_0} - \log C_\theta. \tag{15}$$

With $\log N_{\text{fiber}} = O(d)$, $\log(1/j_0) = O(1)$, and $\omega_D^{1/D} = O(1)$, the leading term dominates for $\varepsilon \in (0, 2\pi/e)$ and $n$ sufficiently large, giving $\gamma_{\text{NISQ}} = c_0 d \log(1/\varepsilon) - O(d/n) = \Omega(d \log(1/\varepsilon))$. $\square$

### C.2.2. FTQC ANALYSIS: DISCRETE PROGRAM REPRESENTATION

**Setup.** At the logical level, FTQC programs are naturally discrete: they consist of finite sequences over a fixed instruction set $\mathcal{I}$, which may include elementary fault-tolerant gates (Clifford+T), module invocations (QFT, modular arithmetic, amplitude estimation), lattice-surgery operations at the substrate level, and control-flow primitives. Each instruction has discrete arguments: qubit or register indices, module parameters, scheduling annotations. The choice of representation (textual IR

such as QASM, AST-level representations, module-based descriptions as in Section 4.2, or planar quantum ISAs (Beverland et al., 2022)) is immaterial for the argument; what matters is that the program space is a discrete set of finite description length.

Let $\mathcal{A}_d^{\text{FTQC}}$ denote the set of structurally valid program descriptions of length at most $G = \Theta(nd)$ over $\mathcal{I}$. Each position selects one of $|\mathcal{I}|$ instructions and up to $\mathcal{O}(1)$ qubit indices from $[n]$, giving

$$|\mathcal{A}_d^{\text{FTQC}}| \ \leq \ (|\mathcal{I}| \cdot n^{\mathcal{O}(1)})^G \ = \ e^{\Theta(nd \log n)}. \tag{16}$$

Compilation defines a map $f^{\text{FTQC}} : \mathcal{A}_d^{\text{FTQC}} \to SU(2^n)$. We bound the conditional pass rate by counting against the uniform distribution on $\mathcal{A}_d^{\text{FTQC}}$:

$$P_{x \sim \pi_\theta}(\|U(x) - U_{\text{target}}\| \leq \varepsilon) \leq C_\theta \cdot \frac{|\{x \in \mathcal{A}_d^{\text{FTQC}} : \|U(x) - U_{\text{target}}\| \leq \varepsilon\}|}{|\mathcal{A}_d^{\text{FTQC}}|}, \tag{17}$$

where $C_\theta := \|\pi_\theta\|_\infty \cdot |\mathcal{A}_d^{\text{FTQC}}|$ measures deviation from uniform.

**Proposition C.2** (Semantic entropy gap, FTQC). *Under the setup above, for any $U_{target} \in SU(2^n)$ admitting a polynomial-length canonical description and any $\varepsilon \in (0, 1)$,*

$$P_{x \sim \pi_\theta}(\|U(x) - U_{target}\| \leq \varepsilon \mid x \in \mathcal{M}_{struct}) \leq C_\theta \cdot e^{-\gamma_{FTQC} n}, \tag{18}$$

*with $\gamma_{FTQC} = \Omega(d \log n)$.*

*Proof.* By the Solovay–Kitaev theorem (Dawson & Nielsen, 2006) applied to a universal fault-tolerant gate set, $U_{\text{target}}$ admits an $\varepsilon$-approximating program of length $G^* = \mathcal{O}(\text{poly}(n, \log(1/\varepsilon)))$. The number of length-$G$ programs that $\varepsilon$-approximate $U_{\text{target}}$ is bounded by the product of (i) the number of $\varepsilon$-equivalent canonical forms of $U_{\text{target}}$, which is $\text{poly}(n, \log(1/\varepsilon))$ for polynomially-describable targets, and (ii) the number of no-op insertions and rewrites of length $G - G^*$, which is at most $|\mathcal{I}|^{G-G^*} \cdot \text{poly}(n)$. The numerator of Equation (17) therefore satisfies

$$|\{x : \|U(x) - U_{\text{target}}\| \leq \varepsilon\}| \leq \text{poly}(n, \log(1/\varepsilon)) \cdot |\mathcal{I}|^{G-G^*}. \tag{19}$$

Dividing by Equation (16), the qubit-index factor $n^{\Theta(G)}$ in the denominator dominates:

$$P(\cdot) \leq C_\theta \cdot \frac{\text{poly}(n, \log(1/\varepsilon))}{n^{\Theta(G)}} = C_\theta \cdot e^{-\Theta(G \log n) + \mathcal{O}(\log n)}, \tag{20}$$

giving $\gamma_{\text{FTQC}} = \Omega(d \log n)$ after substituting $G = \Theta(nd)$. $\qquad\square$

**Remarks (Jointly for both NISQ and FTQC).**

**(i) Where $C_\theta$'s boundedness comes from.** The bound holds pointwise for any $U_{\text{target}}$ in both regimes, but the constant $C_\theta$ distinguishes interesting from uninteresting cases. For targets in (or close to) the model's training distribution — canonical algorithmic primitives such as QFT, modular arithmetic, or Grover diffusion operators — the model may concentrate probability on the correct region of $\mathcal{A}_d$, making $C_\theta$ large enough to cancel the $e^{-\gamma n}$ factor. This is consistent with the empirical success of code-completion systems on textbook quantum algorithms (Dupuis et al., 2024; Vishwakarma et al., 2024): the model is not violating the proposition, it is operating in a regime where $C_\theta$ encodes implicit memorization. The proposition bites for *out-of-distribution* targets, where $C_\theta$ is bounded by definition of generalization. The validity gap is therefore not about whether quantum circuits can in principle implement $U_{\text{target}}$ (universal approximation guarantees this), but about whether a polynomial-time sampler can find the correct circuit without already having seen the answer.

**(ii) Regime independence of the qualitative bound.** Both $\gamma_{\text{NISQ}} = \Omega(d \log(1/\varepsilon))$ and $\gamma_{\text{FTQC}} = \Omega(d \log n)$ yield exponential decay in $n$ for any polynomial depth $d$. The exponents differ in their dependence on tolerance ($1/\varepsilon$ for continuous parameters, $n$ for discrete qubit-index selections) but coincide qualitatively: in both settings, the conditional pass rate is $e^{-\Omega(n)}$ for fixed depth and precision, regardless of the physical substrate.

**(iii) The exponent gets stronger with depth.** A deeper ansatz (larger $d$) gives a *tighter* bound in both regimes. This is the counterintuitive consequence of dimension counting / enumeration: a richer ansatz contains more candidate circuits per

dimension of $SU(2^n)$, so the fraction landing near any specific target is smaller, not larger. The bound is not in conflict with universal approximation theorems for quantum circuits, which establish that sufficiently deep circuits can approximate any unitary, the propositions concern the *fraction* of such approximators among all candidates, not their existence.

**(iv) Regime independence of the validity gap.** The argument is independent of whether the qubits are physical (NISQ) or encoded (FTQC). The dimension parameter $n$ in Equations (11) and (18) refers to the logical qubit count exposed to the algorithm designer. Hardware progress changes the substrate but not the target Hilbert-space dimension, and the validity gap is a property of the target space, not the implementation.

**(v) NISQ pathologies.** The assumption (R1) $j_0 = \Omega(1)$ in Proposition C.1 excludes ansatzes in the *barren-plateau* regime, where $j_0 = e^{-\Omega(n)}$ and the bound degrades. Barren plateaus are a well-known failure mode of variational quantum algorithms (McClean et al., 2018; Cerezo et al., 2021); they signal that the parameter space is exponentially poorly conditioned for gradient-based optimization, which independently rules out efficient synthesis. The proposition therefore captures the relevant regime: either the ansatz is well-conditioned and our bound applies, or the ansatz is in a barren plateau and synthesis is already known to fail for optimization-theoretic reasons.

**(vi) Generic vs. symmetric ansatzes.** (R1)–(R2) hold for "generic" parameterized ansatzes but may fail for ansatzes with built-in symmetries (equivariant ansatzes preserving problem-specific group actions, quantum convolutional networks). For such structured ansatzes the proposition can still apply under restriction to a symmetry-quotient parameter space, but the constants and effective dimension change. We treat this as outside the scope of this paper: structured ansatzes are precisely where verifier-centric agents (Section 4) succeed by exploiting structure, and the validity gap is most acute for unstructured synthesis problems where (R1)–(R2) hold by default.

## D. Property-preserving Rewrite

We illustrate Level 1–3 reasoning by a single depth-reduction move for the Cuccaro ripple-carry adder (Cuccaro et al., 2004).

**Step 1 (Level 1: program over modules).** Write the $n$-bit ripple-carry adder in terms of black-box modules MAJ and UMA and we get the initial program $x_0$:

$$
\begin{aligned}
&\text{ADDER}(A, B, c_0, z): \\
&\quad \text{MAJ}(A_0, B_0, c_0) && \text{Line 1} \\
&\quad \textbf{for } i = 1 \text{ to } n-1: \ \text{MAJ}(A_i, B_i, A_{i-1}) && \text{Line 2} \\
&\quad \text{CNOT}(A_{n-1}, z) \\
&\quad \textbf{for } i = n-1 \text{ to } 1: \ \text{UMA}(A_i, B_i, A_{i-1}) \\
&\quad \text{UMA}(A_0, B_0, c_0)
\end{aligned}
$$

$\text{MAJ}(x, y, z) = (yz \oplus xy \oplus xz, y \oplus x, z \oplus x)$ propagates the carry forward, and $\text{UMA}(x, y, z) = (x \oplus yz, x \oplus y \oplus z \oplus yz, x \oplus y \oplus yz)$ uncomputes the carry while producing sum bits.

**Step 2 (Level 2: choose gate decompositions).** The MAJ gate can be inlined substituted with the following decomposition:

$$
\begin{aligned}
&\text{MAJ}(a, b, c): \\
&\quad \text{CNOT}(a, b); \ \text{CNOT}(a, c); \ \text{TOFFOLI}(c, b, a).
\end{aligned}
$$

Substituting this into Line 1 and Line 2 of the adder structure $x_0$ yields a new program $x_1$:

$$
\begin{aligned}
&\text{ADDER}(A, B, c_0, z): \\
&\quad \text{CNOT}(A_0, B_0); && \leftarrow l_1 \\
&\quad \text{CNOT}(A_0, c_0); \ \text{TOFFOLI}(c_0, B_0, A_0). \\
&\quad \textbf{for } i = 1 \text{ to } n-1: \\
&\qquad \text{CNOT}(A_i, B_i); && \leftarrow l_2 \\
&\qquad \text{CNOT}(A_i, A_{i-1}); \ \text{TOFFOLI}(A_{i-1}, B_i, A_i) \\
&\quad \cdots
\end{aligned}
$$

where the remainder of $x_1$ marked with "$\cdots$" is identical to $x_0$.

**Step 3 (Level 3: expose and apply a parallelization rewrite).** Observe in $x_1$ an opportunity for parallelization shown as two lines marked as $l_1$ and $l_2$ respectively. This invokes a property-preserving rewrite step $a_1 = \text{EDIT}[\rho, \alpha]$ with $\rho = \text{Parallelize}$ and $\alpha = (l_1, l_2)$. A concrete manifestation of such rewrite is shown in Figure 5c with the CNOT gate in the (red) dotted box. The program $x_2$ after the rewrite reads as the following:

$$\text{ADDER}(A, B, c_0, z):$$
$$\textbf{for } i = 0 \text{ to } n - 1 : \text{CNOT}(A_i, B_i);$$
$$\text{CNOT}(A_0, c_0); \ \text{TOFFOLI}(c_0, B_0, A_0).$$
$$\textbf{for } i = 1 \text{ to } n - 1 :$$
$$\text{CNOT}(A_i, A_{i-1}); \ \text{TOFFOLI}(A_{i-1}, B_i, A_i)$$
$$\dots$$

From observing $x_2$, one sees that the rewrite collapses the lines $l_1$ and $l_2$ into a single line (or layer), and therefore simplifies the program.

