# OpenReview forum: "Position: Quantum Program Generation Must Prioritize Validity Over Probabilistic Scaling"
_ICML.cc/2026/Position_Paper_Track — ICML 2026 Position Paper Track regular_

### Official Review · Reviewer_U6iz · 2026-02-14

**Significance:** 3
**Argument Clarity:** 3
**Rating:** 5
**Confidence:** 4

**Questions:**

LLM has been proven effective (e.g., AlphaProof) in formal theorem proofs (e.g., Lean), which also require rigorous reasoning. The key difference is that theorem proofs have efficient formal verifiers that can check each step of reasoning on the fly. This leads to the question of whether the problem lies in the lack of mature verification tools in the quantum realm, rather than the failure of the scaling hypothesis itself.

**Alternative Views Section:**

Yes

**Compliance With Llm Reviewing Policy A Conservative:**

Affirmed.

**Discussion Potential:**

3

**Final Justification:**

After reading the authors’ rebuttal, my concerns have been adequately addressed. While I still encourage the authors to further highlight connections between quantum research and other prominent fields, I am now comfortable supporting acceptance of the paper.

**Paper Summary:**

This paper provides a clear argument that quantum program generation and other similar scientific domains with hard physical and mathematical constraints expose a fundamental syntax–semantics gap, where applying probabilistic scaling of large models can be entirely counter-productive. The exponential sparsity of functionally correct outputs makes post-selection intractable, so reliability must come from a constructively verified system.

**Position:**

Yes

**Position In Title:**

Yes

**Related Work:**

2

**Strengths And Weaknesses:**

Strengths:

1. Clearly present that the success in current systems, e.g., Qiskit Code Assistant, stems from domain-specific verification loops or curated human-verified datasets other than raw parameter scaling.

2. Thorough defense and analyses in Sec. 5.

3. Provide constructive suggestions, e.g., a verifier-centric approach and trace-based datasets for teaching the models to learn the laws of physics.

Weaknesses:

1. LLM has been proven effective (e.g., AlphaProof) in formal theorem proofs (e.g., Lean), which also require rigorous reasoning. The key difference is that theorem proofs have efficient formal verifiers that can check each step of reasoning on the fly. This leads to the question of whether the problem lies in the lack of mature verification tools in the quantum realm, rather than the failure of the scaling hypothesis itself.

2. Some intermediate paths, e.g., curriculum learning or weakly supervised learning, can also be discussed in the paper.

**Support:**

3

---

> ### Author Rebuttal · Authors · 2026-03-29
>
> Thank you for your very positive feedback and for highlighting the constructive aspects of our verifier-centric proposal. We are pleased that our analysis of the "syntax–semantics gap" aligns with your assessment of the current limitations in generative quantum synthesis.
>
> ---
>
> **On whether the core problem is a failure of scaling or a lack of mature verification tools**
>
> You raised a question about whether the problem in quantum generation is truly a failure of scaling, or simply a lack of mature verification tools, analogous to the maturity of Lean/AlphaProof in formal theorem proving. Our view is that the problem lies in a combination of both.
>
> There is a distinct lack of mature verification tools that prove the correctness of a given circuit implementation of a quantum algorithm, though in some (rare) cases end-to-end implementations are formally verified (for instance: https://arxiv.org/abs/2204.07112). Part of this is that it takes a long time to study and implement a quantum algorithm end-to-end even without formal verification, and formalising quantum algorithms down to the implementation details (such as the Figure 5 described in the paper) is still a frontier research topic.
>
> On the scaling law front, currently available data on quantum programs are far more scarce than classical programs. Even though some of the datasets so far are curated with verification in mind and we mentioned them in the paper, it remains open as to how well existing LLMs can generalise based on these datasets on tasks of quantum generation that demand genuine insight that cannot be directly gleaned from the datasets.
>
> ---
>
> **On intermediate paths: curriculum learning and weakly supervised learning**
>
> Your suggestion about discussing intermediate paths such as curriculum learning or weakly supervised learning is well-taken. We believe there are multiple ways these intermediate paths can be approached in our setting.
>
> *Autoformalization as a stepping stone.* One idea is to use autoformalization to create labelling functions that help with verifying a large number of quantum program candidates. Although the task of discovering genuinely novel quantum programs is hard and, as we argue, likely beyond the probabilistic scaling due to complex constraints, the task of recognising valid quantum programs is a much easier task and can indeed benefit from performance improvements driven by the scaling law. We therefore speculate that LLM-powered autoformalization may be a stepping stone for improving the performance of our verifier-centric agent scheme.
>
> *Curriculum-like training from exemplary sequences.* Another idea is closer to curriculum learning, where we take the data of an exemplary action sequence and use it to train the agents by playing the sequence from the beginning. This naturally constitutes a "curriculum" since in an exemplary sequence, the agent likely starts from the "easy" steps such as Level 1 modular decomposition, and progresses towards more nuanced Level 3 optimisation of gate sequences. This curriculum-like approach could be one way to leverage historical data to iteratively improve the agent in our scheme. A similar idea has been proposed in the reinforcement learning setting already (https://arxiv.org/pdf/2402.05808), albeit in the reverse order, the main idea nonetheless centres on the common theme of curriculum learning.
>
> ---
>
> We are very grateful for your insightful comments, which provided a perspective that will undoubtedly strengthen our manuscript. We would be pleased to incorporate these reflections on verification maturity, scaling, and curriculum learning strategies into the final version of the paper.

---

> > ### Author Rebuttal · Reviewer_U6iz · 2026-04-01
> >
> > I appreciate the authors’ comprehensive response. I also hope to see more discussion that combines quantum research with other prominent fields, providing broader insights for a wider readership.

---

> > > ### Author Response · Authors · 2026-04-02
> > >
> > > We certainly look forward to deepen the subjects at the intersection between quantum and other fields, and in that sense appreciate the recognition and encouragement from the reviewer for our efforts.

---

### Official Review · Reviewer_V97g · 2026-03-05

**Significance:** 3
**Argument Clarity:** 3
**Rating:** 5
**Confidence:** 3

**Questions:**

1. **What exactly is “quantum program generation” in your scope?**
   Are you mainly talking about inventing new circuits from scratch, or transforming/optimizing/compiling existing circuits to fit hardware constraints?

2. **What do you mean by “valid,” and which notion matters most for your argument?**
   Is validity “unitary,” “runs on a device,” “matches a target circuit/function,” or “meets an error/noise budget”?

3. **Can you walk through one concrete failure example?**
   A small case where a model output looks plausible but is physically/semantically wrong i.e. what is wrong, and what verifier/check would detect it?

4. **In simple terms, why doesn’t “generate many candidates then filter” work?**
   Is the bottleneck mainly that you need too many samples, or that checking each candidate is too expensive or both?

5. **What would a verifier-centric agent look like step-by-step in a minimal prototype?**
   What tools are called in the loop, what constraints are enforced during decoding, and what would you use as a first benchmark to demonstrate progress?

**Alternative Views Section:**

Yes

**Compliance With Llm Reviewing Policy A Conservative:**

Affirmed.

**Discussion Potential:**

3

**Final Justification:**

After reading the paper and the rebuttal, I’m still positive overall. I thought the submission had a clear and timely position, and I liked that it was trying to push on an important question rather than just present a narrow technical result. My main concerns in the review were about scope, what exactly “validity” meant, and whether the verifier-centric alternative was concrete enough.

The rebuttal helped on those points. In particular, the clarification that the focus is on structured synthesis made the scope much clearer, and the concrete example made the argument easier to evaluate. So overall, the rebuttal addressed my main concerns and made me more comfortable with my final recommendation to accept:)!

**Paper Summary:**

The paper takes a clear stance that “just scale up probabilistic code generation” is the wrong paradigm for *generic* quantum program/circuit synthesis. **I’m not a quantum person**, but my understanding is: unlike natural language or many classical coding tasks, quantum programs have hard physical/mathematical validity constraints (e.g., unitarity, reversibility, and hardware/topology constraints), so fluent-looking outputs can still be fundamentally invalid. The authors argue this creates a growing syntax–semantics gap: small local mistakes can globally break interference/entanglement, making copilot-style generation unreliable.

They further claim that “generate-then-filter” won’t scale because valid circuits become exponentially rare as qubit count grows, while full functional verification is itself exponential in the worst case. As a result, they call for a shift from human-centric copilots to **verifier-centric agents** that build constraints and verification into the generation process (e.g., constrained decoding/masks, hierarchical abstractions, and datasets built from solver/verifier-guided traces).

**Position:**

Yes

**Position In Title:**

Yes

**Related Work:**

3

**Strengths And Weaknesses:**

I found the paper easy to follow and very direct about what it’s trying to argue. The central claim that fluent probabilistic generation is a poor fit for *generic* quantum synthesis because validity is governed by hard constraints rather than surface syntax comes through clearly and is supported by a coherent chain of reasoning. Even as a non-quantum person, I thought the “syntax vs. semantics” framing was effective: it communicates why small local mistakes can have global consequences in quantum programs, which helps justify the skepticism toward copilot-style generation.

I also appreciated that the paper is not purely negative. It offers a concrete alternative direction verifier-centric agents which I fully believe in and sketches several mechanisms (constraint-aware decoding, hierarchical abstraction levels, trace-based supervision) that readers can imagine turning into a research agenda.

Since I’m not an expert, I occasionally found the scope a bit hard to pin down. “Quantum program generation” seems to cover a wide range of tasks, and I wasn’t always sure which ones the authors mean by *generic* synthesis versus more structured or restricted settings. Clarifying the intended scope (and which regimes the argument is meant to apply to most strongly) would help readers like me understand how broadly the claims should be interpreted.

Relatedly, a few of the strongest phrases (e.g., “data poisoning,” “post-hoc filtering is intractable/bankrupt”) felt more like high-level conclusions than fully grounded takeaways, because they aren’t quantified much in the paper. Even a small amount of extra concreteness like tight definitions, a simple illustrative calculation, or brief references to observed failure rates in existing tools would make the critique harder to dismiss. Finally, while the verifier-centric direction is compelling, the proposal stays fairly abstract; I would have benefited from a more concrete picture of what “verification-in-the-loop” looks like end-to-end (what gets checked at each stage, what a “trace” contains, and where approximate checks are acceptable).

**Support:**

4

---

> ### Author Rebuttal · Authors · 2026-03-29
>
> Thank you for your review and we are delighted that our framing of the "syntax-semantics gap" resonated with you.
>
> To clarify the item you mentioned regarding scope, our work focuses on what can be called a "structured synthesis" rather than an unconstrained generic synthesis. While generic synthesis seeks to design circuits from scratch at the gate level, our approach assumes a pre-existing global structure (e.g. the MAJ and UMA modules shown in Figure 5). This allows us to enforce constraints at the module level, which is significantly more efficient than searching an unbounded gate-space.
>
> We propose to revise Section 3 with a quantitative analysis demonstrating why the reliance on "generate-then-filter" fails as system scale increases:
>
> To quantify the intractability of the "generate-then-filter" paradigm, let $p$ be the probability that a randomly generated $n$-qubit circuit is functionally correct. Based on the concentration of the Haar measure on $SU(2^n)$, the probability of success decays exponentially as $p \approx 2^{-\gamma n}$, where $\gamma > 0$ is a constant representing the sparsity of the valid solution space. Consequently, the expected number of trials required to produce one functionally valid design is $E[S] = 1/p \approx 2^{\gamma n}$. For a modest 50-qubit circuit, assuming a realistic $\gamma = 1.0$, this requirement amounts to $2^{50}$ attempts. Given that the cost of each functional verification step (e.g., exact state-vector simulation) scales as $O(2^n)$, the total expected computational cost follows $E[C] \propto 2^{\gamma n} \cdot 2^n = 2^{(\gamma + 1)n}$. This leads to a compound exponential barrier that is termed the "exponential wall", which cannot be overcome by simply scaling model parameters, rendering post-hoc filtering computationally bankrupt for general-purpose synthesis.
>
> Answers to the specific questions:
>
> 1. See the comment above regarding "structured synthesis".
>
> 2. Validity in our context is mainly concerned with implementing the specified mathematical operation correctly. For instance if the circuit is specified to be an in-place adder, it needs to correctly implement the map from |x, y, 0> to |x, x+y> for x and y being n-bit strings.
>
> 3. Example 1 in the Appendix shows a failure example of a three-line quantum program:
>
> ```
> CNOT(y, x) # ‘‘x = y’’
>
> CNOT(x, temp) # ‘‘temp = x’’
>
> CNOT(y, x) # uncompute ‘‘x = y’’
> ```
>
> This looks syntactically plausible - each operation is a valid quantum gate that obeys the constraint of quantum physics i.e. unitarity. It also embeds the intent for performing a joint operation (CNOT) between x and y, followed by storing the resulting value of x into temp before reversing the effect of the joint operation by applying it to x and y again. However, such intent is only correctly realized when x and y are in classical states |0> or |1>. When the variable y is in a quantum superposition such as |0>+|1> and the x as well as temp bit is in |0>, it can be shown that the temp variable is no longer just a leftover from the joint operation (CNOT) but a bit that is entangled with the input variable y in a state |0>_y |0>_temp + |1>_y |1>_temp. This coupling between the temp and y bits violates the original intent, since we want temp to be the only leftover from the computation with x and y, with x and y reset to their input values at the beginning so that they can perhaps be recycled for other computational tasks.
>
> One way to easily detect this error is by tracking the stabilizer Z acting on bit y throughout the circuit, due to space limitations we will not present it here in detail but we will add it to Example 1 in the Appendix.
>
> 4. Both, due to exponential decay of success probability, and computational cost of verifying the design. But the latter is largely amortized since the initial circuit as well as the tools are verified a priori.
>
> 5. Example 2 in Appendix provides a step-by-step illustration of how a minimal prototype works. We start from an initial Level 1 description of the adder program (the starting point of our "structured synthesis"), followed the agent invoking a tool of choosing the decompositions for each of the MAJ and UMA modules, which leads to a Level 2 description of the adder program (contains detailed gate sequences but the structure still retains the modular form inherited from Level 1). Then the agent invokes EDIT tools to rewrite the gate sequences of the circuit for further optimization. The ripple-carry adder example is a good first benchmark to demonstate progress because it is a fundamental building block that underlies many quantum algorithms (including the famed Shor's algorithm for integer factorization) and because of that, human experts have already spent considerable effort optimizing the circuit design (see for instance Cuccaro et al https://arxiv.org/abs/quant-ph/0410184).

---

> > ### Author Rebuttal · Reviewer_V97g · 2026-04-01
> >
> > Thanks for the detailed rebuttal. It cleared up my main questions about the scope, what “validity” means, and what a verifier-centric alternative could look like in practice. The structured-synthesis clarification and the concrete example were especially helpful.
> >
> > Overall, this addressed my main concerns and made the contribution much clearer.

---

> > > ### Author Response · Authors · 2026-04-02
> > >
> > > We appreciate the confirmation and acknowledgement from the reviewer, and look forward to updating our manuscript according to the proposed revisions.

---

### Official Review · Reviewer_8K53 · 2026-03-16

**Significance:** 3
**Argument Clarity:** 2
**Rating:** 5
**Confidence:** 3

**Questions:**

- Sec. 4.1: For constrained-decoding-based approaches, there is always a trade-off between expressiveness and test-time overhead. Could the authors discuss this trade-off in the context of the proposed verifier-centric paradigm?
- Sec. 2.3: Not a question, but a comment—beyond training, there is also inverse scaling at test time: https://arxiv.org/abs/2507.14417.

**Alternative Views Section:**

Yes

**Compliance With Llm Reviewing Policy A Conservative:**

Affirmed.

**Discussion Potential:**

3

**Final Justification:**

All my concerns have been resolved. I appreciate the authors' clarification on the difference between quantum program generation and the classic one.

**Paper Summary:**

As opposed to post-selection, this paper proposes using verification-aware architectures (verified training and verified test-time generation) for quantum program generation. In particular, the authors argue that quantum program generation differs from traditional program generation because errors may not be localized, and partially correct quantum programs may not be as valuable as traditional programs. Additionally, the cost for post-selection could grow exponentially.

**Position:**

Yes

**Position In Title:**

Yes

**Related Work:**

3

**Strengths And Weaknesses:**

Pros
- Well-written paper, easy to follow, with illustrative examples
- I especially appreciate the attribution of the success of 'generative then filter' to localized errors, which might not apply to quantum program generation.

Cons
- The proposed direction appears to focus mainly on test-time verification (e.g., constrained decoding), while some training problems (e.g., 'Structural Poisoning from Unverified Corpora') raised by the authors do not appear to be clearly addressed. If training uses only correct quantum programs, does the model need natural language instruction-following capability? Or will the model suffer from limited training data diversity?
- It would be beneficial to identify initial milestones and benchmarks to measure progress on the proposed verifier-centric quantum program generation.

**Support:**

3

---

> ### Author Rebuttal · Authors · 2026-03-29
>
> Thank you for your thoughtful and constructive review. We are encouraged that you recognize the fundamental breakdown of the "generate-then-filter" paradigm in quantum program generation and support our call for verifier-centric architectures. Below, we address your concerns and questions point by point:
>
> > The proposed direction appears to focus mainly on test-time verification (e.g., constrained decoding), while some training problems (e.g., 'Structural Poisoning from Unverified Corpora') raised by the authors do not appear to be clearly addressed.
>
> Your concern regarding the scope of our proposal and the potential for a loss of instruction-following capability is central to our work. To clarify, our framework is designed for both training and inference phases, beyond merely test-time verification.
>
> Regarding the 'Structural Poisoning' we identified: we view current public repositories as 'poisoned' because they encourage the model to learn the distribution of human error rather than the underlying physical laws. To mitigate this, we propose using our verifier-centric agents to perform trajectory searches through the syntax space. The verifier acts as a teacher, providing immediate validity feedback and optimizing metrics, which allows us to curate a high-fidelity dataset of verified circuit trajectories.
>
> Regarding diversity, we propose to distinguish between 'volume' and 'combinatorial diversity.' As noted by Lopes et al. (2017, link: https://dl.acm.org/doi/10.1145/3133908), massive datasets often suffer from high code duplication, meaning that human-written code occupies a relatively small 'sliver' of the combinatorial space of valid programs. By applying verifier-centric generation, we can generate a strictly more diverse set of valid, functionally correct circuits than what is available via scraping, as we are no longer limited to existing (and often redundant) human solutions.
>
> > It would be beneficial to identify initial milestones and benchmarks to measure progress on the proposed verifier-centric quantum program generation.
>
> Thank you for your valuable feedback regarding the need for concrete milestones and benchmarks. From our perspective, the current field lacks a standardized "north star" for measuring progress in verifier-centric quantum program generation. Standard LLM benchmarks (like pass@1) are insufficient because they reward syntactic fluency and become prohibitively inefficient under precise physical constraints due to the quantum nature of the program. To address this issue, the verifier-centric paradigm that we propose (Figure 5) ensures validity by construction (see V(a,s) in 4.1), allowing benchmarks to focus on more advanced aspects such as optimization efficiency and reasoning trajectories. In this framework, success is measured by the agent’s ability to minimize circuit cost through the fewest number of verifier-validated edit steps. With validity built in, our architecture isolates and highlights the agent's ability to reason about circuit optimization, allowing us to truly evaluate the model's intuition about quantum program design.
>
> > Sec. 4.1: For constrained-decoding-based approaches, there is always a trade-off between expressiveness and test-time overhead. Could the authors discuss this trade-off in the context of the proposed verifier-centric paradigm?
>
> While token-level constrained decoding successfully enforces syntactic validity (e.g., M_topo in Sec. 3.1), it is insufficient for deep semantic constraints (M_func in 3,1) where validity is non-local and dependent on the entire circuit's unitary evolution, see Example 1 in Appendix. In contrast, the verifier-centric agent (Figure 4) treats the circuit design as an iterative trajectory search in a Markov Decision Process (MDP) instead of a sequence of tokens. In this sense, the performance-overhead tradeoff manifests differently. While token-level constrained decoding incurs overhead at every step, our verifier-centric framework trades high-frequency, token-level parsing for constant-time semantic-level verification. Consequently, the computational overhead is effectively amortized across the agent's decision trajectory, allowing us to prune vast sub-trees of the search space that would otherwise require exhaustive post-selection.
>
> > Sec. 2.3: Not a question, but a comment—beyond training, there is also inverse scaling at test time: https://arxiv.org/abs/2507.14417.
>
> Thank you for sharing this reference. This paper about test-time inverse scaling is a strong validation for our position. In this context, the verifier-centric architecture acts as a "guardrail" against exactly these failure modes. In the "natural overthinking" scenarios described in your referenced paper, models drift into unfocused exploration. By contrast, our constructive verification protocols (Section 4.1) enforce an admissible action set. This prevents the model from wandering into the "distracted" reasoning paths that produce inverse scaling.

---

> > ### Author Rebuttal · Reviewer_8K53 · 2026-04-02
> >
> > Thanks for the detailed response. All my concerns have been resolved.

---

> > > ### Author Response · Authors · 2026-04-03
> > >
> > > Thanks very much for your positive support and insightful questions.

---

### Decision · Program_Chairs · 2026-04-30

**Decision:**

Accept (regular)

**Comment:**

The reviewers all agreed on a high score, agreed that the paper presents an important argument, agreed that the paper was well written and clear (particularly so for a "quantum" paper), and appreciated the authors attempts to provide concrete mechanisms for supporting their position and claims, including, i.e. module-level synthesis.